# DNA Methylation Dynamics in the Female Germline and Maternal-Effect Mutations That Disrupt Genomic Imprinting

**DOI:** 10.3390/genes12081214

**Published:** 2021-08-06

**Authors:** Zahra Anvar, Imen Chakchouk, Hannah Demond, Momal Sharif, Gavin Kelsey, Ignatia B. Van den Veyver

**Affiliations:** 1Department of Obstetrics and Gynecology, Baylor College of Medicine, Houston, TX 77030, USA; Zahra.Anvar@bcm.edu (Z.A.); imen.chakchouk@bcm.edu (I.C.); momal.sharif@bcm.edu (M.S.); 2Duncan Neurological Research Institute, Texas Children’s Hospital, Houston, TX 77030, USA; 3Epigenetics Programme, Babraham Institute, Cambridge CB22 3AT, UK; hannah.Demond@babraham.ac.uk; 4Centre for Trophoblast Research, University of Cambridge, Cambridge CB2 3EG, UK; 5Department of Molecular and Human Genetics, Baylor College of Medicine, Houston, TX 77030, USA

**Keywords:** oocyte, epigenetics, genomic imprinting, DNA methylation, subcortical maternal complex, embryo arrest, infertility, epimutations

## Abstract

Genomic imprinting is an epigenetic marking process that results in the monoallelic expression of a subset of genes. Many of these ‘imprinted’ genes in mice and humans are involved in embryonic and extraembryonic growth and development, and some have life-long impacts on metabolism. During mammalian development, the genome undergoes waves of (re)programming of DNA methylation and other epigenetic marks. Disturbances in these events can cause imprinting disorders and compromise development. Multi-locus imprinting disturbance (MLID) is a condition by which imprinting defects touch more than one locus. Although most cases with MLID present with clinical features characteristic of one imprinting disorder. Imprinting defects also occur in ‘molar’ pregnancies-which are characterized by highly compromised embryonic development-and in other forms of reproductive compromise presenting clinically as infertility or early pregnancy loss. Pathogenic variants in some of the genes encoding proteins of the subcortical maternal complex (SCMC), a multi-protein complex in the mammalian oocyte, are responsible for a rare subgroup of moles, biparental complete hydatidiform mole (BiCHM), and other adverse reproductive outcomes which have been associated with altered imprinting status of the oocyte, embryo and/or placenta. The finding that defects in a cytoplasmic protein complex could have severe impacts on genomic methylation at critical times in gamete or early embryo development has wider implications beyond these relatively rare disorders. It signifies a potential for adverse maternal physiology, nutrition, or assisted reproduction to cause epigenetic defects at imprinted or other genes. Here, we review key milestones in DNA methylation patterning in the female germline and the embryo focusing on humans. We provide an overview of recent findings regarding DNA methylation deficits causing BiCHM, MLID, and early embryonic arrest. We also summarize identified SCMC mutations with regard to early embryonic arrest, BiCHM, and MLID.

## 1. Introduction

DNA methylation is one of the best-studied epigenetic modifications and plays an essential role in mammalian development [1]. It is thought to be involved in multiple processes including the regulation of gene expression, heterochromatin formation, and genome integrity. DNA methylation is deposited by the de novo DNA methyltransferases. DNA methyltransferase 3A and 3B (DNMT3A and DNMT3B) are the major de novo DNA methyltransferases, and DNMT3L, which does not possess enzymatic activity, stimulates DNMT3A and DNMT3B activity by binding to their catalytic domains [2]. DNA methylation is maintained during replication by the maintenance DNA methyltransferase 1 (DNMT1) in collaboration with ubiquitin-like with PHD and RING finger domains 1 UHRF1 [3,4]. However, DNMT1 also contributes to de novo DNA methylation in mouse oocytes [5].

Sexual reproduction in mammals requires the reprogramming of methylation marks. Nearly all DNA methylation is removed through a comprehensive reprogramming process in primordial germ cells (PGCs), which are the gamete precursors. This step is referred to as erasure (Figure 1). Subsequently, new methylation patterns are re-established in the germ cells in an asymmetrical fashion in the male and female germline, which give rise to highly specialized gametes ready for fertilization [6]. This step is called establishment (Figure 1). Studies in mice have demonstrated that in the male germline, de novo methylation begins prenatally in mitotically arrested prospermatogonia and is completed after birth [7,8]. The mature sperm has a uniform methylation pattern similar to somatic cells: the genome is ubiquitously hypermethylated with the exception of CG-rich regions of the genome, called CpG islands, most of which remain unmethylated [9,10]. In female germ cells, the establishment of methylation takes place after birth in growing oocytes arrested at meiotic prophase I. The mature oocyte in both mice and humans has a comparatively low overall methylation level that is almost exclusively restricted to transcribed gene bodies but about 10% of CpG islands is also hypermethylated [11,12,13,14].

Following fertilization, the genome of the mouse zygote undergoes another reprogramming wave (Figure 1). The timing and mode of reprogramming differ between the parental genomes. Most of the paternal genome is actively demethylated: Demethylation occurs rapidly and has been observed even before pronuclear fusion [15]. In part, it is mediated by Ten-Eleven Translocation 3 (TET3)-driven oxidation of 5-methylcytosine (5mC) to 5-hydroxymethylcytosine (5hmC) [16]. Conversely, maternal-genome methylation is largely protected at this stage and is slowly diluted during DNA replication, mainly because of the absence of DNMT1 activity [15,17]. However, it has been shown in both human and mouse that the maternal genome is subject to both de novo methylation and hydroxylation events [18,19].

In mammals, genomic imprinting is an important consequence of sexual reproduction. Genomic imprinting is an epigenetically regulated process in which DNA methylation marking in a parent-of-origin-specific manner of one allele of a gene or its control element causes monoallelic expression [20]. Many imprinted genes are organized in clusters and each cluster is regulated by an imprinting control region (ICR). An ICR that acquires mono-allelic DNA methylation in the germline serves as a primary imprinting mark and is, thus, referred to as germline differentially methylated region (gDMR). These primary gDMRs escape embryonic reprogramming and their parent-of-origin dependent mono-allelic DNA methylation persist in somatic tissues. This step is called maintenance (Figure 1) [21].

Altered monoallelic expression of imprinted genes caused by genetic or epigenetic defects is associated with imprinting disorders [22]. In other words, any disruption in erasure, establishment or maintenance of imprints can result in an imprinting disorder. For example, the failure to establish imprints during female gametogenesis or maintain them is shown to be associated with BiCHM, which is an aberrant human pregnancy characterized by placental overgrowth and absence of embryonic tissues [23,24]. The results from studies of patients with imprinting disorders suggest that maintenance of gDMR methylation during post-fertilization reprogramming requires the expression of maternal-effect genes in the oocyte and early embryo. Several of these encode proteins of the subcortical maternal complex (SCMC). The SCMC is a multi-protein structure present in the mammalian oocyte and early embryo involved in several important cellular processes in the egg-to-zygote transition. Maternal ablation of some SCMC members impairs embryonic development and is implicated in adverse reproductive outcomes, including pregnancy loss and hydatidiform moles as well as live-born children with multi-locus imprinting disturbances (MLID) [25,26,27,28,29]. In the latter, an imprinting disorder is accompanied by methylation defects at multiple ICRs.

This review investigates the relationship between DNA methylation deficits and abnormal early development in humans, focusing on the female germline. Therefore, we first describe the DNA methylation landscape in human PGCs, oocytes, pre-implantation embryos, and extra-embryonic lineages in detail. Second, by focusing on the role of the SCMC, we address the current understanding of defective DNA methylation reprogramming in relation to three forms of the restricted developmental disorder: MLID, BiCHM, and early embryonic demise. We also summarize SCMC mutations that impair development by causing the above conditions. Although MLID and BiCHM are rare genetic disorders of imprinting, knowledge of their etiology, together with a full understanding of the mechanisms of imprint establishment and maintenance, are key to recognizing the potential that factors, such as adverse maternal physiology, nutrition or other exposures, and assisted reproduction procedures, have for disrupting the fidelity of imprinting and precipitating lifelong epigenetic errors.

## 2. DNA Methylation Programming and Reprogramming in the Female Germline and Early Embryo

### 2.1. Primordial Germ Cells and Oocytes

Human PGCs are specified at approximately two weeks post-fertilization (pf), after which they migrate from the yolk sac wall and colonize the gonadal ridge between 3–5 weeks pf. Here, they undergo substantial proliferation before sexual differentiation [30]. Until week 5, the genomes of PGCs are highly methylated (Figure 1), but must be reprogrammed to give rise to functional gametes. DNA methylation reprogramming at this stage is concomitant with the downregulation of DNMT3A, DNMT3B, and UHRF1, the DNMT1 accessory factor, and the upregulation of TET1 and TET2 [31]. The lowest DNA methylation level in the human genome is achieved after this reprogramming at 10–11 weeks pf. At this stage, global DNA methylation levels were reported to be 8% in male PGCs at 11 weeks and 6% in female PGCs at 10 weeks (Figure 1) [32,33].

The timing of methylation establishment during oogenesis in humans has recently been described by Yan et al. [34]. Single-cell multi-omics sequencing revealed that de novo methylation occurs in growing oocytes, similar to what has been described in the mouse. De novo methylation correlates with chromatin accessibility and transcription [34]. Although the timing and pattern of de novo methylation during human oogenesis may be analogous to those in mice, the roles of DNMTs may differ. In mice, knockout studies have shown that de novo methylation in the oocyte requires DNMT3A and its co-factor DNMT3L, but not DNMT3B [35,36]. Interestingly, transcripts for DNMT3B are expressed 10-fold higher than DNMT3A in human oocytes and DNMT3L is not expressed at all. Therefore, it has been suggested that DNMT3B, or an oocyte-specific isoform of DNMT3B, may replace DNMT3L in human oocytes as a partner for DNMT3A [13,37]. In the Yan et al. study, DNMT1 and DNMT3A transcripts were found in both growing and fully grown oocytes, while DNMT3B and UHRF1 were mostly expressed in mature oocytes [34].

Mature human oocytes (meiotic metaphase II, MII) show an intermediate level of genomic CpG methylation with a mean of 53.8% [13,38]. Like mouse oocytes, the human oocyte has a bimodal DNA methylation pattern, with broad hypermethylated and hypomethylated domains [13,34,39]. The majority of hypermethylated domains correspond to actively transcribed gene bodies and gene body methylation levels correlate with transcription levels. The remainder of the genome remains largely unmethylated, indicating that active transcription may drive DNA methylation in human oocytes in a similar manner as in the mouse [13,34,38,40]. However, unlike in mice, promoter accessibility positively correlates with gene-body methylation in human oocytes. Although comparable numbers of transcripts are detected in mouse and human oocytes, more genes are hypermethylated (>75% methylation) in human oocytes. In addition, a greater number of non-transcribed intergenic regions that become methylated in human growing oocytes have been identified [34]. In humans, follicle growth is an extended process: almost 120 days are required for primordial follicles to reach the preantral follicle stage and a further 85 days to reach the ovulatory follicle stage [41]. The relatively long time over which DNMTs are active has been suggested as a possible cause of methylation of a greater proportion of genes in humans compared to mice [42]. Alternatively, there may be some differences in the chromatin determinants of de novo methylation or expression levels of necessary chromatin factors.

In their exploration of the oocyte and sperm methylome, Okae and colleagues identified differentially methylated regions (DMRs) between male and female gametes (defined as ≥80% methylation present in only one gamete) and described 29,424 regions that are exclusively methylated in the oocyte genome [13]. These oocyte-specific hypermethylated domains localize primarily to intragenic regions and are enriched for CpG islands (CGIs), gDMRs, gene promoters, gene bodies, and transposable elements such as short interspersed nuclear elements (SINEs), the evolutionary younger superfamily of Alu elements, and long interspersed nuclear elements (LINEs) [19,38].

### 2.2. Preimplantation Embryo

Soon after fertilization, the genome undergoes global demethylation (Figure 1). The removal of epigenetic marks inherited from gametes leads to the restoration of developmental potency in the zygote and adjusts the parental epigenomes of the embryo [43]. This global wave of epigenetic reprogramming is part of the egg-to-embryo transition, which also includes zygotic genome activation (ZGA) and the removal of maternal factors. The initial cell divisions and growth of the developing new embryo rely entirely on stored proteins and RNAs that are expressed from the maternal genome in the developing oocyte. The genes from which these RNAs are expressed are referred to as maternal-effect genes as they are transcribed from the maternal genome before fertilization, but their products are essential for the developing embryo until its genome is transcribed. After fertilization, the maternally provided transcripts and proteins gradually degrade while ZGA (which is completed at the four to the eight-cell stage in human embryos and at the two-cell stage in mouse embryos) encompasses the beginning of embryonic transcription, which will gradually take over control of processes, such as cell differentiation and future development [44,45,46].

DNA demethylation in the human preimplantation embryo occurs in a stepwise manner reaching a mean DNA methylation level of 25.7% in blastocyst-stage embryos at five to six days pf [13,19]. This is significantly lower than the median DNA methylation levels in the egg of 54.5% and sperm of 82%, indicating a significant reduction from the highly specialized gametes to the totipotent embryo [13,19]. Generally, the oocyte hypermethylated domains maintain intermediate methylation levels in the blastocyst [47], but they respond differently to global demethylation based on their genomic properties. For example, transposable elements which are hypermethylated in oocytes are drastically demethylated in blastocysts, with the exception of SINE-VNTR-Alu (SVA) and LTR12 subfamilies, which retain high levels of residual methylation throughout the preimplantation period [13,48]. More so than in mouse embryos, highly methylated CGIs retain substantial levels of methylation in human embryos (median of methylation of these CGIs and gDMRs in the blastocyst is 37.5% and 39.2%, respectively) [13]. The maternal genome is less demethylated in humans than in mouse embryos during preimplantation development. Consequently, the global methylation levels of the human blastocyst closely resemble those of the oocyte [47].

Demethylation of the paternal genome is much faster and more profound than that of the maternal genome in human embryos, which is more similar to the same process in the mouse, suggesting that active demethylation is conserved between the two species [38,48]. Accordingly, the residual DNA methylation levels, in either male pronuclei or the paternal genome domains from the 2-cell stage onward are always lower compared to those of the maternal genome [19,38].

In addition to genome-wide demethylation events, limited de novo methylation of active repeat elements during preimplantation development in human embryos has recently been reported [19]. In particular, SINEs, LINEs, and long terminal repeats (LTRs) acquire DNA methylation at two stages: from the early male pronuclear to the mid-pronuclear stage and from the four-cell to the eight-cell stage on the paternal genome. The function of this de novo methylation has not been elucidated, and the majority of de novo methylated sites lose their methylation again after the next cell division. One hypothesis is that de novo methylation of active repeat elements guarantees genome stability by further repressing those elements at a stage where genome integrity is at its greatest risk [19,38,49].

### 2.3. Post-Implantation Embryo

The blastocyst represents the stage at which the first cellular lineages have become apparent, comprising the trophectoderm (TE), which gives rise to the extraembryonic tissues, and the inner cell mass (ICM), which gives rise to embryonic tissues. In humans, the embryo implants into the uterine wall approximately on day seven after fertilization. By the time the blastocyst progresses to the post-implantation stage, the genomes of both embryonic and extraembryonic tissues have undergone massive tissue-specific and stage-specific DNA-remethylation (Figure 1) [50]. In the mouse, DNMT3A/B are responsible for this de novo methylation which is essential for gastrulation and proper cellular differentiation into the three germ layers: Endoderm, mesoderm, and ectoderm [51,52]. However, in mice, the regulation of DNA methylation in the extraembryonic lineages (which give rise to the placenta) is different from that in the embryonic lineages [53]. Human placental villi are composed of trophoblast, which derives from the TE, and mesenchyme, which originates from ICM-derived extraembryonic mesoderm [54]. As mentioned above, during post-implantation development the genome acquires de novo methylation. However, the placenta maintains a general hypomethylated epigenome compared to the embryo [55]. The placental methylome is organized into partially methylated domains (PMD) and highly methylated domains (HMD) [56]. PMDs are large domains covering about 40% of the genome with an average DNA methylation level of 45% (compared to 80% in HMDs) and overall lower transcription levels than the rest of the genome. The placenta is the only somatic tissue with such extensive PMDs, which are stable throughout gestation. Genes enriched within PMDs have low gene-body methylation levels and lower gene expression compared to those within HMD.

## 3. Methylation Patterning of Imprinted Genes

Imprinted genes occupy a small fraction of the genome and are mono-allelically expressed in a parent-of-origin-dependent manner. They fulfill important functions during the development and disruption of their mono-allelic expression results in several different imprinting syndromes, depending on the affected gene(s) [57]. Many imprinted genes are organized in clusters and each cluster is regulated by an ICR [20,58]. Canonical imprinted genes have differential methylation of the ICR that is established in the germline (gDMR) and is protected from the waves of demethylation and re-methylation during early embryonic development. In contrast, non-canonical imprinting is a DNA-methylation-independent imprinting process [59,60]. This mode of imprinting has recently been described in mice, but it is not yet known if it also occurs in humans and its significance also remains unclear. Non-canonical imprinting is set by H3K27me3 in the oocyte, which results in mono-allelic expression of the corresponding genes from the paternal allele in the preimplantation embryo. However, the H3K27me3 mark is lost during preimplantation development and replaced by monoallelic DNA methylation during post-implantation development specifically in the extra-embryonic lineages [60,61]. Further research is needed to understand how this form of imprinting is maintained throughout pre-implantation development, whether the mechanism is conserved in other species and the particular function of placenta-specific imprinting.

The majority of imprinted genes are epigenetically specified in the female germline. Maternal gDMRs are set during the phase of de novo DNA methylation in the growing oocyte along with the remainder of the methylome. While the rest of the genome is subject to reprogramming after fertilization, gDMRs are largely protected and maintain their methylation from the zygote to post-implantation stage [38,62]. In mice, DNA methylation of the methylated allele of ICRs is maintained by ZFP57, a Kruüppel-associated box (KRAB)-containing zinc finger protein. ZFP57 recognizes and binds to the methylated allele of all murine and most human gDMRs and recruits its cofactor KAP1 (also known as TRIM28). The ZFP57/KAP1 complex then recruits other epigenetic modifiers, including SETDB1 and DNMT1, to the gDMRs to protect the methylated allele from demethylation [63,64]. In mice, ZFP57 is continually expressed from the oocyte to the early embryo, and it has been shown that oocyte expression of ZFP57 is required for the proper maintenance of gDMRs [63]. In humans, ZFP57 is expressed only in the early embryo with zygotic genome activation [45,65]. It is unclear if this difference in expression has any effect on function, but loss of function mutations of human ZFP57 disrupt the maintenance of imprinting resulting in MLID [66]. Even so, in these patients, not all imprinted genes are affected suggesting that other complementary factors may play a role. Recent studies identified other zinc finger proteins, ZFP445 and ZNF202, that bind the majority of ICRs and are expressed in oocytes; ZFP445, in particular, appears to be more important for imprinting maintenance in human embryos. In a recent study, for the first time a homozygous ZFP445 variant was found. This pathogenic variant caused Temple syndrome and MLID in the patient [65,67,68].

During the remethylation phase in the post-implantation embryo, the unmethylated allele of canonical imprinted genes is protected against de novo methylation, passing lifelong memory of parental origin into the next generation (Figure 1) [69,70]. In humans, some gDMRs are maintained exclusively in the placenta. This phenomenon is called placenta-specific imprinting and has not yet been observed in the mouse [47,54,71,72,73,74]. Most placenta-specific imprinted genes are transient and retain methylation of maternal origin set in the oocyte [54,71]. Placenta-specific imprinted genes show mono-allelic methylation on the maternal allele in placental villi, cytotrophoblasts, trophoblast, and mesenchyme mostly become unmethylated in somatic tissues [54,71]. The incomplete demethylation of the maternal allele during preimplantation development or incomplete de novo methylation of the paternal allele post-implantation are proposed as mechanisms of placenta-specific imprinting [47].

## 4. Global Loss of Imprinting Results in Hydatidiform Molar Pregnancies

The two waves of DNA methylation reprogramming in the germline and the early embryo are important for normal development. In particular, the establishment and maintenance of imprints have been shown to be crucial for the maintenance of a healthy pregnancy. Therefore, the loss of imprinting results in a variety of developmental abnormalities [21,75]. The most severe form is the hydatidiform mole (HM), which is a gestational abnormality characterized by trophoblast overgrowth and the absence of embryo development [76]. In most cases, HM pregnancies occur sporadically and are the result of an androgenetic embryo that has two paternal genome copies and is lacking the maternal copy. The lack of the maternal copy consequently means that all maternal imprints are missing, while the paternal imprints are fully methylated, which is thought to be the main factor driving the HM phenotype. In mice, androgenetic pregnancies lacking maternal imprints show similarities to the HM phenotype in that they are characterized by trophoblast overgrowth and abnormal development of the embryo proper [77,78]. The lack of imprinting results in the imbalance of imprinted gene expression. In a recent study using bipaternal mice, this imbalance of expression was corrected at seven imprinted loci, which resulted in the birth of live pups, highlighting the importance of mono-allelic expression of imprinted genes for normal pregnancies [79].

In rare cases, HM are recurrent, and in most such instances, this coincides with a biparental genome. These are termed biparental complete hydatidiform mole (BiCHM). The majority of BiCHM pregnancies have been associated with mutations in the maternal-effect genes *NLRP7* (~75%) and *KHDC3**L* (~5–10%) [25]. Recently, a patient with a *PADI6* mutation was identified [80]. In contrast to androgenetic HMs, BiCHM has a maternal copy of the genome. Their phenotype is associated with widespread loss of methylation at (almost) all maternal gDMRs in patients with disease variants in *NLRP7* and *KHDC3L* [22,81,82,83,84]. Given that only the maternal and not paternal gDMRs appear to be affected, it was suggested that the loss of methylation originates in the oocyte. Indeed, a recent study assessing DNA methylation in oocytes of a patient with a *KHDC3L* mutation showed that global DNA methylation establishment in the oocyte was impaired, including but not limited to the maternal gDMRs [24]. While the majority of the methylome recovered post-implantation, as assessed in the molar tissue, thus suggesting normal de novo methylation in the embryo, the maternal gDMRs were not rescued [24]. However, this has only been shown for a single patient with a *KHDC3L* mutation and does not exclude the possibility that in other patients BiCHM pregnancies may be a result of failed gDMR maintenance or a combination of defects in the oocyte and preimplantation embryo. This is supported by patients in which *NLRP7* mutations have been associated with MLID in which paternal gDMRs were also affected [28].

## 5. Molar Pregnancies Indicate a Role for the Subcortical Maternal Complex in Ensuring Imprinting

Although the evidence described above implicates NLRP7 and KHDC3L in the establishment of maternal gDMRs in the oocyte, the mechanism of how this may occur remains unexplained. NLRP7 and KHDC3L are both parts of the same oocyte multi-protein complex, the subcortical maternal complex (SCMC) [85]. The SCMC was first described in mice and regulates several essential cellular processes during the egg-to-embryo transition, such as spindle assembly, chromosome alignment, and symmetric cell division in cleavage-stage embryos [86,87]. The SCMC has since also been detected in other mammalian species, including humans [85,88]. In humans, seven genes have, so far, been described to encode proteins of the SCMC: *NLRP5* (*MATER*), *OOEP* (*FLOPED*), *TLE6*, *NLRP2*, *NLRP7*, *KHDC3L* (*C6ORF221*), and *PADI6* [85,89,90,91].

The SCMC is localized in the subcortical region of the cytoplasm just below the cell membrane or cortex of the oocyte and persists throughout preimplantation development until the blastocyst stage, where it is excluded from regions with cell-cell contact. The cytoplasmic localization of the SCMC makes a potential role in imprinting regulation all the more intriguing. As discussed above, mutations in *NLRP7*, *KHDC3L*, and, potentially, *PADI6* are associated with BiCHM pregnancies. Other genes, including *NLRP5*, *NLRP2*, *NLRP7*, *PADI6*, and *OOEP*, have been associated with MLID and miscarriages [22,27,28,92]. MLID is thought to occur from failure of gDMR maintenance in preimplantation embryos because, in contrast to BiCHM, only a variable subset of gDMRs is affected in MLID. The cause of the high frequency of miscarriages in some women with SCMC mutations is unknown [22,27,28]. One possibility is that the number of imprinted genes affected varies between offspring, and only milder cases develop to term. Another gene frequently mutated in MLID is *ZFP57* [66] which, together with *DNMT1* and *TRIM28*, encodes part of the DNA methylation maintenance machinery of gDMRs during preimplantation development [93]. How the SCMC members impact the maintenance machinery remains to be resolved. One possibility is that the SCMC functions as a regulator of cellular organization and through that can regulate the localization of proteins involved in DNA methylation, such as DNMTs. Indeed, knockout of *Nlrp2* in mice disrupts the subcellular localization of DNMT1, but not DNMT3A [94]. Immunofluorescence showed that DNMT1, which was enriched in the cortex together with other SCMC proteins in control oocytes and preimplantation embryos, had a more diffuse cytoplasmic rather than cortical localization in maternal knockout zygotes [94]. This would suggest an involvement of NLRP2 in DNA methylation maintenance, which fits with the association of NLRP2 with MLID in humans [28,92]. Mid-gestation embryos and neonates from *Nlrp2*-deficient oocytes did indeed show small alterations in methylation at imprinted gDMRs [94]. *Nlrp2* is so far the only mouse gene in which a link between the SCMC and imprinting regulation has been made. It will be interesting to assess the role of *Nlrp2* and other SCMC genes in imprinting in mice more comprehensively, to improve our understanding of conditions such as BiCHM and MLID.

Studying SCMC function in humans is challenging, as the genes are exclusively expressed in oocytes and human material is extremely rare, even more so from patients with pathogenic variants. Furthermore, not all SCMC members are conserved in mammals, as some, such as the NLRPs, belong to rapidly evolving gene families [95]. For example, *NLRP7*, the major gene involved in BiCHM, is not found in the mouse, making the use of animal models difficult. Mouse knockout studies have shown that the SCMC proteins are tightly regulated. Except for *Khdc3* (*Filia*), the mouse orthologue of *KHDC3L*, the knockout of any one of the SCMC genes destabilizes the entire complex and results in dispersing protein localization of the other members [86,96,97]. These knockout studies also showed that ablation of one of the SCMC proteins usually results in embryo arrest at the zygote or early cleavage stages [29,86,98]. This very early embryo arrest is not linked to imprinting, as oocyte-specific deletion of *Dnmt3a* or *Dnmt3l* in mice, which results in complete loss of maternal imprinting, does not cause any phenotype until later in development [35,99].

The discrepancy between human and mouse studies may partly be caused by under ascertainment of human cases. Mosaicism in patients may affect the diagnosis of MLID [21]. Furthermore, pathogenic genetic variants that are compatible with life in the case of MLID or with pregnancy in the case of BiCHM are more likely to be detected than those that cause early preimplantation embryo arrest. There is growing evidence supporting that mutations in genes encoding SCMC members may be associated with early embryo arrest independent of imprinting disorders [100,101,102,103,104]. It has been proposed that there might be a causal link between the SCMC, DNA methylation, and genome integrity, as imprinting aberrations have been associated with aneuploidies in the embryos of patients with SCMC mutations [21,22,28,75]. A role for the SCMC in ploidy is also supported by a mouse study, in which maternal ablation of *Khdc3* caused abnormal spindle assembly, chromosome misalignment, and spindle assembly checkpoint inactivation during the early embryo cleavage stages, resulting in increased aneuploidy rates [96]. A combination of imprinting defects and aneuploidy may therefore be at the core of explaining miscarriages often observed in patients with SCMC mutations.

## 6. Pathogenic Variants Identified within Human SCMC Genes

Numerous families and singletons with early embryonic lethality, MLID, and BiCHM have been studied to find the causative genes for these conditions. DNA from probands and family members have been subjected to whole-genome or whole-exome sequencing, and variants have been classified and annotated. Rare homozygous and compound heterozygous variants have been identified in *NLRP2*, *NLRP5*, *NLRP7*, *KHDC3L*, *TLE6*, *PADI*, *OOEP*, *UHRF1*, and *ZAR1*. In most cases, the segregation in familial presentations was confirmed. Reported variants in genes that encode SCMC proteins and associated clinical features have been compiled in the Table 1.

DAPIN (Domain in Apoptosis and Interferon response) domain. Protein arginine deiminase (PAD), Beckwith-Wiedemann syndrome (BWS), Silver-Russell Syndrome (SRS), transient neonatal diabetes mellitus (TNDM), partial HM (PHM). Functional effects of the detected variations were predicted from SIFT, PolyPhen and CADD Phred (CADD Phred score column is provided in the Appendix A). The Minor Allele Frequency (MAF) of the potential causal variants was found on The Genome Aggregation Database (gnomAD). 

## 7. Conclusions

In this review, we discussed how clinical syndromes, and in particular, BiCHM, which result from imprinting disturbances. Up to 50% of imprinting errors are caused by primary epimutations that alter DNA modifications without altering the DNA sequence. Epimutations are usually the result of post-fertilization random errors in imprinting marks, and thus, are not commonly inherited. Environmental factors, such as nutrition, physiological disturbances, such as metabolic disorders, or exposures to agents, such as endocrine disruptors may cause primary epimutations. In addition, there has been a long-standing concerns that some procedures in assisted reproductive technologies (ART) could induce epimutations. Other imprinting errors result from genetic changes, including chromosomal rearrangements, mutations (in imprinted genes or epigenetic regulators) and uniparental disomies. Here, we highlighted imprinting errors arising from mutations in epigenetic regulators such as ZFP57, as well as from maternal effect mutations in genes that encode proteins of the enigmatic SCMC. The outcomes of such mutations range from serious compromise in developmental competence of the embryo, infertility, to MLID.

To improve molecular diagnosis and clinical management of these, and related, human reproductive disorders, it will be important to improve our understanding of the causes and origin of imprinting errors. This is particularly true for maternal-effect mutations in genes that encode proteins of the SCMC, for which the mechanisms leading to imprinting errors remain obscure. As we discussed in this review, the SCMC is involved in a multitude of functions important in the oocyte and preimplantation embryo and mutations in components of this complex cause serious imprinting errors. More studies are required to disentangle the different functions of the SCMC in humans. Due to the difficulties in accessing human material, SCMC knock-out mouse models could be informative to investigate the underlying mechanisms, but it remains to be seen whether SCMC mutations in the mouse induce DNA methylation defects akin to those in BiCHM or MLID. However, these disorders demonstrate that major epigenetic defects can arise at crucial times of methylation programming and reprogramming events due to defects in factors-the SCMC-that themselves are not epigenetic regulators. This may reveal the importance of as yet undiscovered cellular processes that ensure the activity, cofactor availability or sub-cellular localization of the epigenetic machinery required for DNA methylation establishment or maintenance events. This might also identify a new domain of vulnerability of epigenetic control in the critical preimplantation period that could be sensitive to aspects of adverse maternal physiology or ART procedures.

## Figures and Tables

**Figure 1 genes-12-01214-f001:**
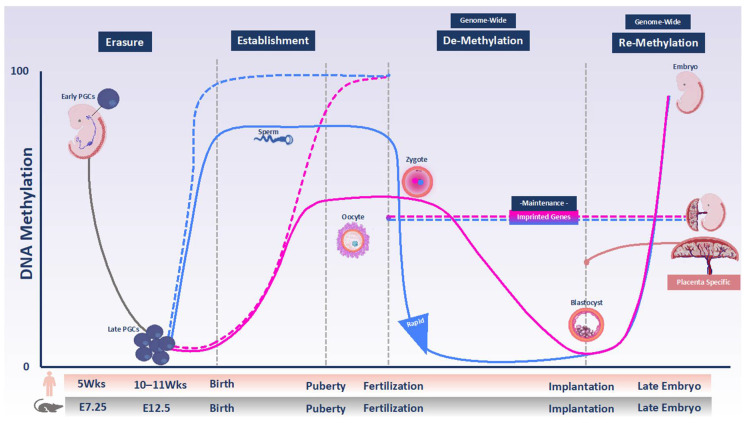
DNA methylation programming and reprogramming during development in humans and mice. DNA methylation is globally erased in primordial germ cells (PGCs) derived from epiblast during their proliferation and migration to the genital ridges (solid gray line). Therefore, de novo DNA methylation subsequent takes place on a largely blank slate during male and female gametogenesis. In the male, new methylation is established from soon after demethylation and is almost completed by the time of birth (solid blue line). In females, there is no gain of methylation until after birth. Growing oocytes arrested in the first meiosis prophase gain methylation between birth and puberty, as well as in adult life (solid pink line). Maternally and paternally imprinted DMRs become differentially methylated in gametes over the same time window (dashed pink and blue lines, respectively). After fertilization, the parental genomes undergo genome-wide demethylation, which does not include imprinted DMRs (dashed combined pink and blue lines). The timing and extent of demethylation are different for the two parental genomes. The paternal genome is rapidly demethylated in part via TET proteins activity (solid blue line). The maternal genome is protected against TET activity and undergoes passive demethylation following DNA replication (solid pink line). By the time of implantation, the genome–except for imprinted DMRs–undergoes re-methylation events that are necessary for cell-lineage determination (combined red and blue lines). Placenta-specific imprinting that is present only in humans is shown with the solid coral orange line.

**Table 1 genes-12-01214-t001:** Summary of familial and singleton variants within SCMC genes causing early embryonic lethality, MLID, and BiCHM.

Gene	Family	hg19 Position	GenBank	cDNA Mutation	Protein Mutation	Mutation Effect	gnomAD_Exomeall MAF	gnomAD_Genomeall MAF	SIFT	Polyphen	Inheritance	Domain/Exon	Pregnancy Outcomes	Country	Ref
*NALP7*	MoLb1	Chr19: 55452298	NM_206828	IVS3+1G>A	2 splicing isoforms: -inclusion of the first 4 bp of intron 3 between exons 3 and 4, addition of two aa followed by a stop codon -exclusion of exon 3	Splicing mutation (Splice donor)	0.00E + 00	0.00E + 00	NA	NA	Autosomal recessive (Homozygous)	Intron 3	Recurrent hydatidiform moles	Lebanon	[25]
	MoPa61	Chr19: 55445856		IVS7+1G>A	inclusion of the entire intron 7	Splicing mutation	5.17E-05	NA	NA	NA	Autosomal recessive (Homozygous)	Intron 7	Complete hydatidiform mole, spontaneous abortion (7–20 weeks)	Pakistan	
	MoGe2	Chr19: 55449464		2077C>T	p.Arg693Trp	Missense mutation	2.74E-04	6.69E-04	NA	NA	Autosomal recessive (Homozygous)	Exon 5	Complete hydatidiform mole	Germany	
	MoIn68	Chr19: 55449463		2078G>C	p.Arg693Pro	Missense mutation	4.77E-05	NA	tolerated (0.07)	benign (0.056)	Autosomal recessive (Homozygous)	Exon 5	Complete hydatidiform mole	India	
	MoIn69-2	Chr19: 55441939		c.2738A>G	p.Asn913Ser	Missense mutation	1.35E-04	7.33E-04	deleterious (0)	probably_damaging (0.991)	Autosomal recessive (Compound heterozygous)	Exon 5	Complete hydatidiform mole and invasive mole	India	
		Chr19: 55449463		c.2078G>C	p.Arg693Pro	Missense mutation	4.77E-05	NA	tolerated (0.07)	benign (0.056)		Exon 9			
*NLRP7*	Family 6	Chr19: 55447768	NM_001127255.1	c.2161C>T	p.Arg721Trp	Missense mutations	5.97E-05	NA	NA	NA	Compound Heterozygous	Exon 7	BWS–MLID	Germany	[28]
		Chr19: 55445006		c. 2573T>C	p. Ile858Thr		7.16E-05	6.37E-05	deleterious (0)	benign (0.351)		Exon 8			
	Family 7	Chr19: 55451438		c.749T>G	p.Phe250Cys	Missense mutations	4.57E-04	4.14E-04	deleterious (0)	possibly_ damaging (0.88)	Compound Heterozygous (Mother)	NACHT domain	BWS and TNDM		
		Chr19: 55451083		c. 1104T>G	p.Ile368Met		4.84E-04	5.49E-04	NA	NA	Heterozygous in Proband	Exon 4			
	Family 8	Chr19: 55447773	NM_206828.2	c.2156C>T	p.Ala719Val	Missense mutation	1.05E-03	1.05E-03	deleterious (0.01)	probably_damaging (0.963)	Heterozygous (Mother and Proband)	Exon 6	SRS	UK	[28]
														China	[91]
														Italy	[105]
	Patient 1 and 2	Chr19: 55449463	NM_001127255.1	c. 2078G>C	p.Arg693Pro	Missense mutation	4.77E-05	NA	tolerated (0.07)	benign (0.056)	Autosomal recessive	Exon 5	Complete hydatidiform moles	UK	[84]
	Patient 3	Chr19: 55449184_55454887 del		c.-39-1769_2129+ 228del		Deletion of exons 2-5	NA	NA	NA	NA	Autosomal recessive	5′UTR			
	Patient 4	Chr19: 55449523		c.2018C>G	p.Ser673Ter	Nonsense mutation	3.98E-06	NA	NA	NA	Compound Heterozygous	Exon 5			
		Chr19: 55447768		c.2161C>T	p.Arg721Trp	Missense mutation	5.97E-05	NA	NA	NA		Exon 6			
	Family E	Chr19: 55451235_55451248	NM_206828.3	c.939_952 dup 14	p.Tyr318Cys fsTer7	Frameshift mutation	2.39E-05	1.27E-04	NA	NA	Compound Heterozygous	Exon 4	Familial biparental hydatidiform mole	UK	[106]
		Chr19: 55449511		c.2030delT	p.Leu677Pro fsTer6	Mutations	NA	NA	NA	NA		Exon 5			
	Family N	Chr19: 55449523		c.2018C>G	p.Ser673Ter	Nonsense mutation	3.98E-06	NA	NA	NA	Autosomal recessive	Exon 5		Pakistan	
	Family J	chr19: 55452305		c.346A>T	p.Lys116Ter	Nonsense mutation	NA	NA	NA	NA	Autosomal recessive	Exon 3		Pakistan	
	Family K	Chr19: 55449463		c.2078G>C	p.Arg693Pro	Missense mutation	4.77E-05	NA	tolerated (0.07)	benign (0.056)	Autosomal recessive	Exon 5		Pakistan	
	Family L	Chr19: 55451049		c.1138G>C	p.Gly380Arg	Missense mutation	6.66E-04	NA	NA	NA	Heterozygous	Exon 4		Pakistan	
	Singleton 1	chr19: 55445994		c.2334G>A	p.Trp778Ter	Nonsense mutation	NA	NA	tolerated (0.16)	probably_damaging (0.95)	Autosomal recessive	Exon 7	Molar pregnancy	Pakistan	
	Singleton 2	Chr19: 55450731		c.1456dupG	p.Glu486Gly fsTer42	Frameshift mutation	NA	NA	NA	NA	Autosomal recessive	Exon 4		Punjabi	
	Singleton 4	Chr19: 55450994		c.1193T>G	p.Leu398Arg	Missense mutation	3.48E-05	NA	tolerated (0.16)	probably_damaging (0.95)	Autosomal recessive	Exon 4		Pakistan	
	Singleton 5	Chr19: 55449463		c.2078G>C	p.Arg693Pro	Missense mutation	4.77E-05	NA	tolerated (0.07)	benign (0.056)	Autosomal recessive	Exon 5		Pakistan	
	Singleton 6	Chr19: 55452802		c.277+1G>C		Splicing mutation	NA	NA	NA	NA	Autosomal recessive	Intron 2		Pakistan	
	Singleton 7	Chr19: 55450994		c.1193T>G	p.Leu398Arg	Missense mutation	3.48E-05	NA	tolerated (0.16)	probably_damaging (0.95)	Autosomal recessive	Exon 4		Pakistan	
	MoCh76	Chr19: 55452356	NM_206828.2	c.295G>T	p.Glu99Ter	Nonsense mutation	NA	NA	NA	NA	Compound Heterozygous	Exon 3	BiCHM	China	[91]
		Chr19: 55449571		c. 1970A>T	p. Asp657Val	Missense mutation	NA	NA	NA	NA		Exon 5			
	Ch29	Chr19: 55447764		c.2165A>G	p.Asp722Gly	Missense mutation	3.98E-06	NA	deleterious (0.05)	possibly_ damaging (0.574)	Autosomal recessive	Exon 6	BiCHM		
	Ch77	Chr19: 55450893		c.1294C>T	p.Arg432Ter	Nonsense mutations	3.61E-05	3.19E-05	NA	NA	Compound Heterozygous	Exon 4	CHM		
		Chr19: 55445108		c.2471+1G>A	p.Leu825Ter		NA	NA	NA	NA		Exon 7			
	Ch101	Chr19: 55449440		c. 2101C>T	p.Arg701Cys	Missense mutation	1.99E-05	NA	tolerated (1)	benign (0.018)	Compound Heterozygous	Exon 5	BiCHM		
		Chr19: 55449463		c.2078G>A	p.Arg693Gln		7.95E-06	3.19E-05	tolerated (0.07)	benign (0.056)					
	MoCh195	Chr19: 32436314_55448111 del1218		c.2130-312_2300+ 1737del1218			NA	NA	NA	NA		Exon 6	CHM		
	MoCh200	Chr19: 55450487_55450562 del76		c.1625_1700 del76	p.Met542Thr fsTer2	Frameshift mutation	NA	NA	NA	NA	Compound Heterozygous	Exon 4	HM		
		Chr19: 55445108		c. 2471+1G>A	p.Leu825Ter	Nonsense mutation	NA	NA	NA	NA		Exon 7			
	MoCh293	Chr19: 55450893		c.1294C>T	p.Arg432Ter	Nonsense mutation	3.61E-05	3.19E-05	NA	NA	Compound Heterozygous	Exon 4	HM		
		Chr19: 55447773		c.2156C>T	p.Ala719Val	Missense mutation	1.05E-03	1.05E-03	deleterious (0.01)	probably_damaging (0.963)		Exon 6			
	MoCh73	Chr19: 55451050		c.1137G>C	p.Lys379Asn	Missense mutation	5.01E-03	6.08E-03	NA	NA	Heterozygous	Exon 4	CHM		
	MoCh71	Chr19: 55452829		c.251G>A	p.Cys84Tyr	Missense mutation	4.53E-04	3.20E-04	tolerated (0.05)	benign (0.079)	Heterozygous	Exon 2	AnCHM		
	MoCh193	Chr19: 55451050		c.1137G>C	p.Lys379Asn	Missense mutation	5.01E-03	6.08E-03	NA	NA	Heterozygous	Exon 4	HM		
	MoCh190	Chr19: 55445860		c.2468T>A	p.Leu823Ter	Nonsense mutation	NA	NA	NA	NA	Heterozygous	Exon 7	AnCHM		
	MoCh71	Chr19: 55452829	NM_001127255.1	c.251G>A	p.Cys84Tyr	Missense Mutation	4.53E-04	3.20E-04	tolerated (0.05)	benign (0.079)	Heterozygous	Exon 2	CHM, PHM (with no family history of moles)	China	[105]
	MoIt96	Chr19: 55451720		c.467G>A	p.Arg156Gln	Missense mutation	7.25E-03	8.09E-03	tolerated (0.23)	benign (0.125)	Heterozygous	Exon 4	HM (with no family history of moles)	Italia	
	MoCh73	Chr19: 55451050		c.1137G>C	p.Lys379Asn	Missense mutation	5.01E-03	6.08E-03	NA	NA		Exon 4		China	
	MoCa57	Chr19: 55450991		c.1196G>A	p.Cys399Tyr	Missense mutations	4.72E-04	2.87E-04	deleterious (0)	probably_damaging (1)	Compound Heterozygous	Exon 4	CHM/IM (with no family history of moles)	Morocco and Algeria	
		Chr19: 55450727		c.1460G>A	p.Gly487Glu		5.31E-02	1.29E-01	tolerated (0.12)	benign (0.094)		Exon 4			
	MoCa88	Chr19: 55450655		c.1532A>G	p.Lys511Arg	Missense mutation	1.33E-02	2.91E-02	deleterious (0.01)	possibly_ damaging (0.701)	Heterozygous	Exon 4	Recurrent spontaneous abortions, 2 twins Hashimoto disease (with no family history of moles)	Morocco and UK	
	Ch101	Chr19: 55449440		c.2101C>T	p.Arg701Cys	Missense mutations	1.99E-05	NA	tolerated (1)	benign (0.018)	Compound Heterozygous	Exon 5	CHM (with no family history of moles)	China	
		Chr19: 55449463		c.2078G>A	p.Arg693Gln		7.95E-06	3.19E-05	tolerated (0.07)	benign (0.056)					
	MoCa94	Chr19: 55447773		c.2156C>T	p.Ala719Val	Missense mutation	1.05E-03	1.05E-03	deleterious (0.01)	probably_damaging (0.963)	Heterozygous	Exon 6	PHM (with no family history of moles)	Italy	
	Ch29	Chr19: 55447764		c.2165A>G	p.Asp722Gly	Missense mutation	3.98E-06	NA	deleterious (0.05)	possibly_ damaging (0.574)	Autosomal recessive (Homozygous)	Exon 6	PHM, BiCHM, CHM (with no family history of moles)	China	
	MoUs99	Chr19: 55447681		c.2248C>G	p.Leu750Val	Missense mutation	5.29E-04	9.56E-05	NA	NA		Exon 5	PHM, CHM, HM (Familial recurrent HMs)	Mexico	
	Ch77	Chr19: 55450893		c.1294C>T	p.Arg432Ter	Nonsense mutations	3.61E-05	3.19E-05	NA	NA	Compound Heterozygous	Exon 4	CHM	China	
		Chr19: 55445108		c.2471+1 G>A	p.Leu825Ter		NA	NA	NA	NA		Intron 7			
	MoFr101	Chr19: 55439063		c.2891T>C	p.Leu964Pro	Missense mutation	NA	NA	deleterious (0)	probably_damaging (1)	Autosomal recessive (Homozygous)	Exon 10	PHM	France	
*NLRP5*	Family 1	Chr19: 56544020	NM_153447.4	c.2320T>C	p.Cys774Arg	Missense mutation	4.02E-06	NA	deleterious (0)	probably_damaging (0.997)	Compound Heterozygous (mother and 2 probands)	LRR domain	Proband 1 with SRS-MLID (heterozygous c.2320T > C)	UK	[27]
		Chr19: 56539263		c.1664G>T	p.Gly555Val	Missense mutation	NA	NA	deleterious (0)	probably_damaging (0.944)		NACHT domain	Proband 2 with BWS-MLID (heterozygous c.1664G > T)		
	Family 2	Chr19: 56544053		c.2353C>T	p.Gln785Ter	Nonsense mutation	8.43E-05	NA	NA	NA	Compound Heterozygous in the mother and proband 1. c.2840T > C not inherited by either affected offspring	LRR domain	Proband 1 with BWS–MLID. Proband 2 with a clinically non-specific autism and obesity–MLID	UK	
		Chr19: 56552341		c.2840T>C	p.Leu947Pro	Missense mutation	2.61E-04	2.23E-04	deleterious (0)	probably_damaging (0.996)					
	Family 3	Chr19: 56515174		c.155T>C	p.Met52Thr	Missense mutation	8.02E-06	NA	tolerated (0.07)	benign (0.007)	Compound Heterozygous	DAPIN domain (N-terminal effector)	Proband with BWS–MLID	UK	
		Chr19: 56515245		c.226G>C	p.Glu76Gln		8.02E-06	NA	deleterious (0)	probably_damaging (0.999)					
	Family 4	Chr19: 56538755		c.1156_1158 dupCCT	p.386dupPro	Missense mutation	4.03E-06	NA	NA	NA	Heterozygous in the mother but not inherited in either twin	NACHT domain	Proband (one of discordant monozygotic pair) was SRS–MLID	Germany	
	Family 5	Chr19: 56539298		c.1699A>G	p.Met567Val	Missense mutation	4.42E-05	NA	tolerated (0.11)	benign (0.017)		NACHT domain	MLID, presenting with atypical	UK	
													clinical features of BWS and Prader–Willi syndrome		
	Family 6	Chr19: 56515311		c.292C>T	p.Gln98Ter	Nonsense mutation	NA	NA	NA	NA	Compound Heterozygous	Pyrin	Recurrent early embryonic arrest	China	[104]
		Chr19: 56539680		c.2081C>T	p.Thr694Ile	Missense mutation	NA	NA	deleterious (0)	probably_damaging (0.973)		LRR			
	Family 7	Chr19: 56538465		c.866G>A	p.Gly289Glu	Missense mutation	NA	NA	deleterious (0)	probably_damaging (1)	Compound Heterozygous	NACHT			
		Chr19: 56569626		c.3320C>T	p.Thr1107Ile	Missense mutation	NA	3.19E-05	deleterious (0)	probably_damaging (0.993)		LRR			
	Family 1	Chr19: 56538660		c.1061C>T	p.Pro354Leu	Missense mutation	1.21E-05	3.19E-05	deleterious (0.03)	probably_damaging (0.999)	Autosomal recessive	NACHT	Recurrent early embryonic arrest	China	[107]
*NLRP2*	Family 1	Chr19: 55494543	NM_017852.4	c.1479_1480 delAG	p.Arg493Ser fsTer32	Frameshift mutation	7.56E-05	NA	NA	NA	Autosomal recessive (Homozygous Mother), Heterozygous in both probands	LRR domain	MLID	Germany	[28]
	Family										Autosomal recessive consanguineous family		Proband with BWS–MLID	Pakistan	[92]
	Family 2	Chr19: 55497553		c.2237delA	p.Asn746Thr fsTer4	Frameshift mutation	3.98E-06	NA	NA	NA	Heterozygous mother and proband	Exon 8	Proband with SRS	Germany	Family previously reported in [108] and [109]
	Family 3	Chr19: 55505788		c.2860_2861 delTG	p.Cys954Gln fsTer18	Frameshift mutation	NA	NA	NA	NA	Heterozygous mother	Exon 11/LRR domain	Proband 47, XXY, Symmetrical growth restriction and developmental delay	Germany	[28]
	Family 4	Chr19: 55485901		c.314C>T	p.Pro105Leu	Missense mutation	2.79E-05	NA	tolerated (0.15)	possibly_ damaging (0.604)	Heterozygous mother	Exon 3	TNDM		[28]
	Family 5	Chr19: 55494951		c.1885T>C	p.Ser629Pro	Missense mutations	1.01E-03	1.12E-03	deleterious (0)	probably_damaging (0.959)	Compound Heterozygous (Mother and Proband)	Exon 6	SRS	UK	[28]
		Chr19: 55501424		c. 2401G>A	p. Ala801Thr		9.17E-03	1.27E-02	tolerated (0.51)	benign (0.097)		Exon 9			
	Family 1	Chr19: 55495027	NM_017852.5	c.1961C>A	p.Ser654Ter	Nonsense mutation	NA	NA	NA	NA	Autosomal recessive	Exon 6	MLID	China	[104]
	Family 2	Chr19: 55493839		c.773T>C	p.Phe258Ser	Missense mutation	3.98E-06	NA	deleterious (0)	probably_damaging (0.993)	Compound Heterozygous	NACHT			
		Chr19: 55497571		c.2254C>T	p.Arg752Ter	Nonsense mutation	3.98E-06	NA	NA	NA		Exon 9			
	Family 3	Chr19: 55493591		c.525G>C	p.Trp175Cys	Missense mutation	NA	NA	tolerated (0.06)	probably_damaging (0.979)	Compound Heterozygous	Exon 6			
		Chr19: 55501876		c.2544A>T	p.Glu848Asp	Missense mutation	NA	NA	deleterious (0.01)	probably_damaging (0.994)		LRR			
	Family 4	Chr19: 55493728		c.662C>T	p.Thr221Met	Missense mutation	8.85E-02	9.05E-02	deleterious (0.04)	probably_damaging (0.989)	Compound Heterozygous	NACHT			
		Chr19: 55494913		c.1847A>T	p.Glu616Val	Missense mutation	7.96E-06	NA	deleterious (0.04)	benign (0.405)		Exon8			
	Family 5	Chr19: 55493728		c.662C>T	p.Thr221Met	Missense mutation	8.85E-02	9.05E-02	deleterious (0.04)	probably_damaging (0.989)	Compound Heterozygous	NACHT			
		Chr19: 55494534		c.1469C>T	p.Arg490Cys	Missense mutation	1.28E-04	3.20E-05	deleterious (0.01)	benign (0.03)		Exon7			
*KHDC3L*	Family L	Chr6: 74072455	NM_001017361.3	c.3G>T	p.Met1Ile next available downstream ATG codon lies at residue 14	Loss of start codon	3.98E-06	NA	deleterious (0)	probably_damaging (0.916)	Autosomal recessive (consanguineous family)	Exon1	Familial Biparental Hydatidiform Mole	Pakistan	[26]
	Family T	Chr6: 74072970		c.322_325 delGACT	p.Asp108Ile fsTer30	Frameshift mutation	2.39E-05	NA	NA	NA		Exon 2	Complete Hydatidiform Mole	Tunisia	
	Family W	Chr6: 74072453		c.1A>G	p.Met1Val	Missense mutation	NA	NA	deleterious (0)	probably_damaging (0.916)	Compound Heterozygous	Exon 1	Complete Hydatidiform Mole	Asia	
		Chr6: 74072969		c.322_325 delGACT	p.Asp108Ile fsTer30	Frameshift mutation	2.39E-05	NA	NA	NA		Exon 2			
	Patient D	Chr6: 74072453		c.1A>G	p.Met1Val	Start codon loss	NA	NA	deleterious (0)	probably_damaging (0.916)	Autosomal recessive		BiCHM	Iran	[24]
*TLE6*	Family 1	Chr19: 2993572	NM_001143986.2	c.1529C>A	p.Ser510Tyr	Missense mutation	NA	NA	deleterious (0)	probably_damaging (0.912)	Homozygous in 2 probands	WD40 domain repeats (Cterminal)	Early embryonic Arrest (1,2 and 4 cell stage)	Saudi Arabia	[100]
	Family 2										Homozygous in consanguineous family				
*PADI6*	Family 1	Chr1: 17720537	NM_207421.4	c.1141C>T	p.Gln381Ter	Nonsense mutation	NA	NA	NA	NA	Homozygous in consanguineous family	PAD domain	Early Embryonic Arrest (arrested at the 2- to 4-cell stage)	China	[101]
	Family 2	Chr1: 17727858		c.2009_2010 del	p.Glu670Gly fsTer48	Frameshift mutation	NA	NA	NA	NA	Compound Heterozygous	PAD domain	Early Embryonic Arrest (arrested at the 1- to 2-cell stage)		
		Chr1: 17708541		c.633T>A	p.His211Gln	Missense mutation	3.21E-05	NA	deleterious (0.02)	probably_damaging (0.936)		PAD middle domain			
	Family 3	Chr1: 17722159		c.1618G>A	p.Gly540Arg	Missense mutation	4.08E-06	NA	tolerated (0.05)	benign (0.159)	Compound Heterozygous	PAD domain	Early Embryonic Arrest (arrested between the 2- and 5-cell stages)		
		Chr1: 17718616		c.970C>T	p.Gln324Ter	Nonsense mutation	NA	NA	NA	NA					
	Family	Chr1: 17725285		c.1793A>G	p.Asn598Ser	Missense mutation	NA	NA	tolerated (0.05)	probably_damaging (0.911)	Compound Heterozygous	PAD domain	Recurrent hydatidiform moles (RHM)	China	[80]
		Chr1: 17727894		c.2045 G>A	p. Arg682Gln	Missense mutation	8.03E-06	NA	deleterious (0)	probably_damaging (0.992)					
	Family 1	Chr1: 17718714		c.1067G>A	p.Trp356Ter	Nonsense mutation	NA	NA	NA	NA	Probands mother is Compound Heterozygous	PAD domain (Exon 10)	Beckwith-Wiedemann syndrome with multi-locus imprinting disturbance	Italy	[110]
		Chr1: 17727743		c.1894C>G	p.Pro632Ala	Missense mutation	4.01E-06	NA	deleterious (0)	probably_damaging (1)		PAD domain (Exon 17)			
	Family 2	Chr1: 17721538		c.1429A>G	p.Met477Val	Missense mutation	4.01E-06	NA	tolerated (0.48)	possibly_ damaging (0.452)	Proband’s mother is Compound Heterozygous	PAD domain (Exon 13)			
		Chr1: 17727929		c.2080C>T	p.Pro694Ser		8.05E-06	NA	deleterious (0)	probably_damaging (1)		PAD domain (Exon 17)			
	Family 3	Chr1: 17727855		c.2006delC	p.Thr669Lys fsTer85	Frameshift deletion	NA	NA	NA	NA	Heterozygous	PAD domain (Exon 17)			
*PADI6 (hg38)*	Family 9	Chr1: 17388820	NM_207421.3	c.902G>A	p.Arg301Gln	Missense mutations	NA	NA	deleterious (0)	probably_damaging (1)	Compound Heterozygous (Mother) Proband not tested	Exon 8	SRS		[28]
		Chr1: 17394415		c.1298C>T	p.Pro433Leu		NA	2.63E-05	deleterious (0)	probably_damaging (1)		Exon 11			
	Family 10	Chr1: 17394024		c.1124T>C	p.Leu375Ser	Missense mutations	NA	NA	deleterious (0.01)	probably_damaging (0.915)	Compound Heterozygous (Mother)	Exon 10	BWS–MLID		
		Chr1: 17397091		c.1639G>A	p.Asp547Asn		NA	5.06E-04	tolerated (1)	benign (0.005)	Heterozygous in Proband	Exon 14			
	Family 11	Chr1: 17392197		c.1046A>G	p.Asp349Gly	Missense mutation	NA	NA	tolerated (0.37)	probably_damaging (0.953)	Heterozygous (Mother)	Exon 9	SRS	Germany	
	Family 12	Chr1: 17379985		c.433A>G	p.Lys145Glu	Missense mutation	NA	6.57E-06	deleterious (0.02)	possibly_damaging (0.612)	Heterozygous (Mother)	Exon 4	SRS	Germany	
*OOEP (hg38)*	Family 13	Chr6: 73369684	NM_001080507.2	c.109C>T	p.Arg37Trp	Missense mutation	NA	3.29E-05	deleterious (0.04)	benign (0.135)	Autosomal recessive (Homozygous Mother), Heterozygous proband	Exon 1	TNDM		
*UHRF1 (hg38)*	Family 14	Chr19: 4930782	NM_013282.4	c.514G>A	p.Val172Met	Missense mutation	NA	NA	deleterious (0)	probably_damaging (0.952)	Heterozygous (Mother and Proband)	Exon 3	SRS		
*ZAR1 (hg38)*	Family 15	Chr4: 48492438	NM_175619.2	c.130G>T	p.Glu44Cys	Missense mutation	NA	NA	deleterious (0.01)	possibly_damaging (0.748)	Heterozygous (Mother and Proband)	Exon 1	mild macroglossia, and high birth weight, but no other features of BWS		

The letter E is substituted for “× 10^”. Not Applicable (NA).

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
