# Peer review of "DNA Methylation Dynamics in the Female Germline and Maternal-Effect Mutations That Disrupt Genomic Imprinting"

_genes, 2021, doi:10.3390/genes12081214_

Round 1

Reviewer 1 Report

Anvar and colleagues performed a very great work on this intriguing but also very complex field. The paper is well organized and written. Some paragraph is not very fluent but it’s very hard to expose the complexity of some mechanism described.
Figure 1 is essential in order to help the reader in following the different states of the erasure/ establishment/de-methylation/ re-methylation phases.
I appreciated the realization of the table 1. In the NGS era is very important to have a validated e contextualized report of mutations individuated in the literature. I'm sure the bad layout of the table 1 is due to in progress version of the paper.
If a good English reviser could make the reading more fluent, this will help the reader in appreciating all the information collected in this review.

Author Response

Thank you for your kind comments on our manuscript. We hope that this review article will increase understanding of the DNA methylation landscape in the female germline and related pathologies.

We have edited the manuscript for English language and style and the edited file has been uploaded.

Reviewer 2 Report

This review article described about the mechanisms of DNA methylation in female germlines, and mutations in the genes comprising SCMC and hydatidiform moles and imprinting abnormalities. The reviewer thinks that this article is well-written and should be published. I have a few comments.

1) Regarding Figure 1, which paper is the methylation level of the sperm from birth to puberty based on? The authors need to cite the paper.

2) In Figure 1, it is difficult to distinguish between the solid pink line that indicates methylation of the maternal genome of the fetus and the solid red line that indicates methylation of the placenta. It would be better to change the color.

3) It should be clearly stated whether the discussion from line 82 to 90 is based on human data or mouse data.

4) In relation to lines 290-291, a case of MLID with ZNF445 mutation has been recently published (2021 Kagami et al. Clinical Epigenetics). The authors should be cited this article.

5) How do the authors consider the pathogenicity of hetero-missense variants of PADI6, UHRF1, and ZAR1 genes? Could you evaluate the pathogenicity of these variants using  in silico analysis, frequency analysis, and/or ACMG guideline?

Author Response

1) Thank you for mentioning this point. Methylation levels in human sperm and mouse sperm are reported by Okae and colleagues 2014 and Henckel and colleagues 2012, respectively. Okae et al 2014 was already included among the references. Henckle et al 2012 is now also cited in the appropriate place within the text and added to the reference list (see red highlighted text).

2) Thank you for pointing out this potential point of confusion. We have changed the color of the placenta-specific imprinting line to coral orange.

3) The reviewer’s comment is very important. This paragraph is mainly about mouse data and only the last sentence regarding de novo methylation and hydroxylation is true for the both species. We have clarified this in the revised version.

4) Thank you so much for mentioning this important reference. “Kagami et al 2021” is now added to the references and cited in the appropriate place.

5) Thank you for this comment about the effect of heterozygous missense variant on function. We have expanded the table and provided the required information. Four new columns were added to Table 1, that include gnomAD Exomeall MAF, gnomeAD Genomeall MAF, SIFT, and Polyphen scores. Above information in addition to a column with CADD Phred score has also been uploaded as supplementary information in an excel sheet. 

x) We have reviewed and edited the paper for English grammar and style and the edited file is uploaded.